# Low Psychosine in Krabbe Disease with Onset in Late Infancy: A Case Report

**DOI:** 10.3390/ijns7020028

**Published:** 2021-05-28

**Authors:** Camille S. Corre, Dietrich Matern, Joan E. Pellegrino, Carlos A. Saavedra-Matiz, Joseph J. Orsini, Robert Thompson-Stone

**Affiliations:** 1Department of Neurology, University of Rochester School of Medicine and Dentistry, Rochester, NY 14642, USA; camille_corre@urmc.rochester.edu; 2Biochemical Genetics Laboratory, Department of Laboratory Medicine and Pathology, Mayo Clinic, Rochester, MN 55905, USA; matern@mayo.edu; 3Inherited Metabolic Specialty Center, Department of Pediatrics, Upstate Medical University, Syracuse, NY 13010, USA; pellegrj@upstate.edu; 4NY State Newborn Screening Program, Wadsworth Center, New York State Department of Health, Albany, NY 13010, USA; carlos.saavedra@health.ny.gov (C.A.S.-M.); joseph.orsini@health.ny.gov (J.J.O.)

**Keywords:** Krabbe disease, psychosine, newborn screening

## Abstract

Krabbe disease (KD) is a rare inherited neurodegenerative disorder caused by a deficiency in galactocerebrosidase enzyme activity, which can present in early infancy, requiring an urgent referral for hematopoietic stem cell transplantation, or later in life. Newborn screening (NBS) for KD requires identification and risk-stratification of patients based on laboratory values to predict disease onset in early infancy or later in life. The biomarker psychosine plays a key role in NBS algorithms to ascertain probability of early-onset disease. This report describes a patient who was screened positive for KD in New York State, had a likely pathogenic genotype, and showed markedly reduced enzyme activity but surprisingly low psychosine levels. The patient ultimately developed KD in late infancy, an outcome not clearly predicted by existing NBS algorithms. It remains critical that psychosine levels be evaluated alongside genotype, enzyme activity levels, and the patient’s evolving clinical presentation, ideally in consultation with experts in KD, in order to guide diagnosis and plans for monitoring.

## 1. Introduction

Krabbe disease (KD) is a rare, autosomal recessive neurodegenerative disorder caused by a deficiency in galactocerebrosidase (GALC) enzyme activity. The phenotypic spectrum of KD is broad and extends from the most severe form, early infantile Krabbe disease (EIKD), to manifestations with an onset after 6 months of life through adulthood termed late onset Krabbe disease (LOKD). EIKD has a rapidly progressive onset in the first few months of life, ultimately leading to a devastating loss of neurologic function and death, whereas the course for LOKD is more variable [1,2].

Hematopoietic stem cell transplantation (HSCT) has been shown to favorably modify the disease course in patients with KD, if performed prior to the onset of neurological symptoms [3,4,5]. Given the potential benefit for disease-modifying therapy when diagnosed early, newborn screening (NBS) for KD was introduced in New York State in 2006 and has since expanded to an additional eight states. Although GALC enzyme activity is used as a first-tier screening tool [6,7], experience has shown that this biomarker alone is insufficient to distinguish between EIKD, LOKD, and false-positive screens due to carrier state and/or enzyme-lowering *GALC* polymorphisms (pseudodeficiency alleles). Early protocols introduced *GALC* genotyping as a second-tier test, but this testing continued to be burdened by poor specificity due to the high prevalence of variants of unknown significance [7,8].

The biomarker psychosine (Psy) has emerged as a means of risk-stratifying patients found with reduced GALC enzyme activity because it accumulates in patients with EIKD [9,10,11,12]. Therefore, it was developed as a second-tier NBS test performed when GALC activity is found to be reduced [13,14,15,16,17]. The assay was improved to increase its accuracy at low concentrations, leading to the realization that intermediate elevation could better help identify patients at risk for LOKD [9,17]. However, the exact concentrations of Psy in blood that indicate increased risk for LOKD have not been fully established, which complicates the clinician’s ability to organize an appropriate follow-up plan. Through consensus meetings, a group of experts in KD screening, diagnosis, and treatment, the KD NBS Council, has agreed upon the following risk categories using the current assay of Psy in DBS: (1) normal (<2 nmol/L); (2) intermediate (≥2 and <10 nmol/L), which indicates risk of LOKD; and (3) high (≥10 nmol/L), which is consistent with EIKD. However, evidence of the sensitivity of those categorizations is currently lacking [18]. Here, we present a case of KD with onset in late infancy and surprisingly low Psy levels that challenges our understanding of the interplay between genotype, biomarkers, and clinical presentation in the diagnosis of KD.

## 2. Case Report

The patient first presented at 2 weeks of life to pediatric genetics for follow-up of a NBS result presumed positive for KD. The NBS sample was collected at two days of life and results were reported by the NY screening program on the ninth day of life, showing galactocerebrosidase activity of 0.30 μmol/L/h (5.2% of daily mean), a Psy level of 1.2 nmol/L, and compound heterozygosity for two *GALC* variants, considered likely pathogenic: p.Arg396Trp and p.Tyr567Ser along with an enzyme-lowering polymorphism, p.Ile562Thr. His mother reported a normal pregnancy, full-term delivery, and an uncomplicated birth. There was no family history of Krabbe disease, but one of the patient’s older brothers had possible autism by maternal report and was noted to be a late walker with language delay. He had two older brothers who screened negative for KD, and one older maternal half-brother who screened negative. As per the NY’s NBS protocol, blood was sent for determination of GALC enzyme activity in leukocytes and was shown to be in the “high risk” category (0.07 nmol/h/mg protein with high risk ≤0.2 nmol/h/mg protein). In addition, another DBS was sent for Psy analysis and was found to be 2.6 nmol/L, which at the time was interpreted as normal (cutoff 3 nmol/L) (Figure 1). At this time, the patient was asymptomatic with a reassuring neurological exam. Given these findings, the neonate was felt to not have EIKD and serial follow-up was scheduled given the concern for possible LOKD.

At the next follow-up visit at 3 months of life, parents reported that he had been feeding well but was irritable, crying every 2 h, with poor sleep quality. He was noted to have a head lag on examination. Another DBS was sent for Psy analysis, revealing an approximate Psy concentration of 0.8 nmol/L (the assay’s lower limit of quantitation is 1.5 nmol/L) (Figure 1). He was referred for evaluation for therapy services and qualified for physical therapy. At 7 months of life, the patient presented to genetics clinic with continued irritability including periods of inconsolable crying and frequent upper respiratory tract infections. Developmentally, he had gained satisfactory head control and was reaching for objects, but with hands frequently fisted. He was able to sit in tripod for a few seconds, but he could not sit independently when placed. Repeat DBS testing revealed a Psy concentration of 2.0 nmol/L (Figure 1). He was referred for an MRI and formal neurological evaluation at 10 months of age, but the imaging was not completed at that time. His neurological examination revealed significant truncal hypotonia, lower extremity spasticity, and brisk lower extremity reflexes. He had begun to developmentally regress, losing the ability to roll. He began to develop episodes of breath-holding. His EEG was normal and electromyography was felt to be essentially normal but with mildly reduced recruitment and large amplitude motor units, suggesting a possible early neurogenic process. Repeat measurement of Psy in a DBS was approximately 1.0 nmol/L (Figure 1).

At 12 months of age, the patient was referred to a Leukodystrophy Care Center and was noted to have lost some head control, continued to be significantly irritable, and had worsening dysphagia. He was found to have axial hypotonia with limb spasticity, and he could not reliably reach for or grab objects. Brain MRI was performed and showed mild symmetric non-enhancing T2 hyperintensities involving the bilateral dentate nuclei, corona radiata, and centrum semiovale as well as fusiform enlargement of the bilateral optic nerves (Figure 2). A presumptive diagnosis of late onset KD was made based on the clinical presentation and imaging findings [19]. Full-exome sequencing was performed, confirming the *GALC* genotype established by the NY NBS program, but no additional findings were uncovered. Due to his advanced symptomatic state, the patient was felt to not be a candidate for HSCT.

The patient continued to struggle with spasticity, dysphagia, irritability, and constipation, and he was treated symptomatically. By 26 months of age, his clinical status had deteriorated to the point where his family opted to transition to comfort care measures, and he died peacefully, surrounded by family in his home.

## 3. Discussion

The patient described here presented with a NBS result that showed a likely pathogenic *GALC* genotype, GALC activity in the high-risk category, and psychosine concentrations in DBS that fluctuated between normal and mildly elevated. The p.Arg396Trp and p.Tyr567Ser variants have been identified in patients with EIKD who were either homozygous or compound heterozygous with another pathogenic variant [20,21,22,23,24]. The residue p.Arg396 has been shown to be critical as it directly binds substrate in the active site [25] and p.Arg396Trp has significantly reduced enzyme activity compared to wild-type when transfected in COS 1 cells [26]. p.Tyr567Ser by itself or in *cis* with p.Ile562Thr has detrimental effect on protein folding and secretion affecting trafficking to the lysosomes [27,28]. Because the genotype involving these variants had not been reported in other KD patients and was not observed by the NY NBS program in screen positive infants who ultimately were not affected, it was deemed likely pathogenic.

The Psy values obtained were all interpreted as normal at the time, but reference ranges have since been revised, and concentrations of ≥ 2 nmol/L are now considered abnormal. This combination of results caused confusion regarding risk for KD and proper follow-up. In one previously reported case, a 2-month-old patient who went on to develop infantile KD was screened based on family history and found to have a DBS psychosine level of 2.0 nmol/L [9].

Efforts to include Psy as a second-tier NBS test have pointed toward its ability to differentiate EIKD patients from LOKD patients and unaffected individuals [13,14,17]. This differentiation is critical to providing the most appropriate follow-up and care to individuals at risk. In the literature, EIKD has generally been associated with strikingly elevated Psy levels, generally greater than 10 nmol/L. In comparison, unaffected individuals generally have Psy levels less than 2 nmol/L, and patients at risk for LOKD tend to fall somewhere in between the two values [17,18] (Figure 1).

In our patient, only one Psy level was greater than 2.0 nmol/L (2.6 nmol/L), out of five over the first year of life. Yet the patient developed KD in infancy, a devastating outcome not clearly accounted for in NBS algorithms. As the MRI findings were not completely typical of EIKD, it is possible that this child had a secondary diagnosis, although exome sequencing was performed and did not identify any alternative diagnoses. However, exome sequencing has its limitations as certain variants may go undetected (e.g., gross deletions/duplications/rearrangements, and deep intronic variants). Regardless, we believe that this case illustrates the potential for rare cases in which Psy levels are borderline or normal in patients who will ultimately develop Krabbe disease, even in infancy or early childhood. It is difficult to explain the progression of disease with normal levels of psychosine, though, as elevated psychosine has been used to explain mechanism of disease, with elevated psychosine causing apoptosis of oligodendrocytes and the Schwann cells with disease onset (psychosine hypothesis) [12].

Although Psy is an incredibly informative biomarker, further study is required to assess its value beyond the newborn period and to determine whether DBS is the best specimen type to monitor at-risk patients. DBS can be of variable quality due to potentially improper collections (e.g., problems in obtaining good capillary blood flow, variable technique of specimen collection by heel or finger stick, and an inability to correct for hematocrit), which is why Psy analysis in erythrocytes (expressed as pmol/g hemoglobin) has been suggested [17]. Despite this case, we do endorse the use of Psy as a sole second-tier screening test given the extreme rarity of LOKD presenting with a value of <2.0 nmol/L. However, for any patients to come to clinical attention, it remains imperative to evaluate the entire array of available information including genotype, enzyme activity level, Psy concentration, and most importantly, the patient’s evolving clinical presentation. Additionally, given the complexity of interpreting equivocal NBS and follow-up results, we recommend that a pediatric neurologist or geneticist with specific KD expertise be consulted early on [17,18], most ideally as part of a regional Leukodystrophy Care Center (https://www.huntershope.org/family-care/leukodystrophy-care-network/lcn-care-centers/; accessed on 23 April 2021). The Leukodystrophy Care Network coordinates urgent referrals to these centers and hosts monthly and ad hoc meetings of the KD NBS Council to review cases and assist care providers faced with these difficult decisions. Moving forward, it will be critical to create a comprehensive and accessible database to track laboratory values and clinical follow-up data to inform on the spectrum of possible outcomes.

## 4. Conclusions

Psy levels provide invaluable information as part of NBS for KD and their measurement as a second-tier test has improved the specificity of NBS for KD. While the case presented here has led to the revision of Psy cutoffs in DBS and an attempt to standardize results between laboratories providing Psy testing [29], it is unlikely that 100% sensitivity can be achieved without considerable negative impact on specificity [8,30]. Finally, it cannot be overemphasized that Psy levels must be reviewed as just one of several important factors in determining the diagnosis and follow-up plan of a patient who screens positive. In all cases, clinical and laboratory follow-up of newborns at risk of KD should be pursued rapidly and in consultation and close collaboration with KD experts.

## Figures and Tables

**Figure 1 IJNS-07-00028-f001:**
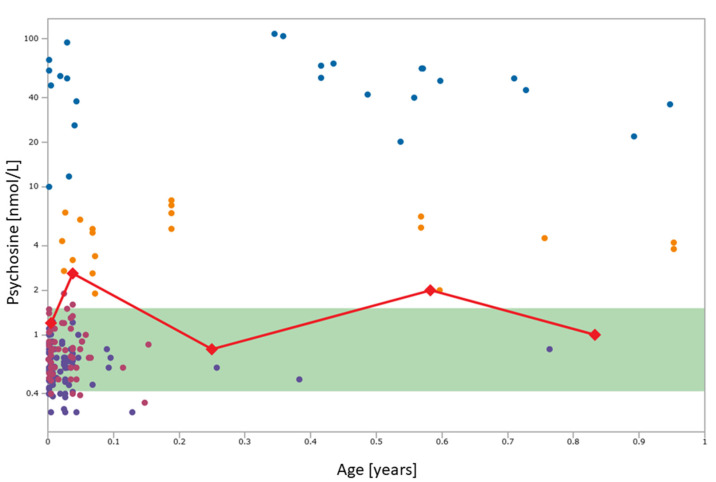
Psychosine concentrations in dried blood spots collected at 2 days old for newborn screening, and at 2 weeks, 3 months, 7 months, and 10 months of age in comparison to the control range (green band), cases with infantile onset Krabbe disease (blue circles; *n* = 25), later onset Krabbe disease variants (orange circles; *n* = 20), *GALC* mutation carriers (magenta circles; *n* = 50), and individuals with GALC activity lowering genotypes (“pseudodeficiency”; purple circles; *n* = 97). Figure created in Collaborative Laboratory Integrated Reports (https://clir.mayo/edu, accessed on 23 April 2021); Psychosine values are shown on a log scale.

**Figure 2 IJNS-07-00028-f002:**
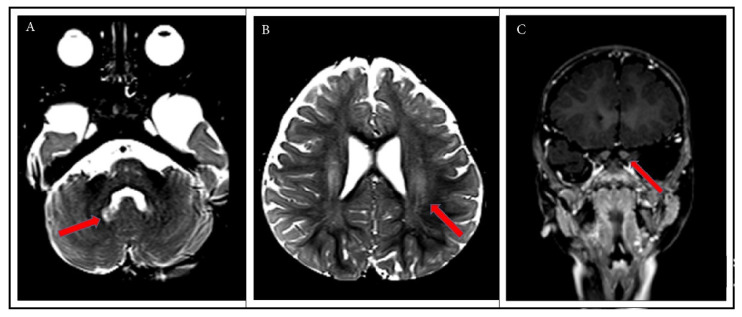
The brain MRI at 12 months of life, showing, as indicated by red arrows, mild symmetric non-enhancing T2 hyperintensities involving the bilateral dentate nuclei (**A**), corona radiata and centrum semiovale (**B**), as well as fusiform enlargement of the bilateral optic nerves (**C**).

## Data Availability

No new data were created or analyzed in this study. Data sharing is not applicable to this article.

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
