# Peer review of "Low Psychosine in Krabbe Disease with Onset in Late Infancy: A Case Report"

_2409-515X, 2021, doi:10.3390/ijns7020028_

Round 1

Reviewer 1 Report

This is a very well written manuscript describing a case report where an (presumptively?) diagnosed with late infantile Krabbe disease patient exhibited low phychosine levels during the first year of life. The manuscript contributed towards our evolving understanding of psychosine levels vs. phenotype of Krabbe disease.

I only have a couple of minor points.

Page 3, line 90, change the <date of submission> with an actual date.

The authors did not address how newborn screening labs should handle Psy second-tier screening results in light of the findings of this case report. Can Psy be used as a sole second-tier screening tool for the screening of LOKDs or molecular second-tier screening is required as well. I understand that this an evolving field of research and the authors may not have all the answers.

Author Response

We thank the reviewer for their useful comments. Below is a point-by-point response:

"Page 3, line 90, change the <date of submission> with an actual date."

  • We fixed that error, now on line 93, the actual date was added.

"The authors did not address how newborn screening labs should handle Psy second-tier screening results in light of the findings of this case report. Can Psy be used as a sole second-tier screening tool for the screening of LOKDs or molecular second-tier screening is required as well. I understand that this an evolving field of research and the authors may not have all the answers."

  • We thank the reviewer for this excellent comment, and we added a sentence to the bottom of page 5, lines 177/178 making a definitive recommendation to maintain Psy as sole second-tier screening tool given the rarity of this situation.

Reviewer 2 Report

Well written case report on a positive Krabbe disease newborn screen case with pathogenic mutations and lower than expected psychosine levels.  The infants clinical course was consistent with Krabbe disease and additional testing with whole exome sequencing was performed to evaluate for other etiologies that could provide an alternative non Krabbe diagnosis for the case.   Authors have extensive experience in newborn screening and Krabbe disease.   Good review of current status of psychosine testing in Krabbe disease and the case report psychosine levels compared with previously diagnosed Krabbe cases.  This case report provides further support for the need to utilize all laboratory  data and clinical course in the diagnosis and follow up planning of newborn screen Krabbe cases. 

Author Response

We thank the reviewers for their time and effort in reviewing our manuscript. There were no specific comments for revision.